# LEARNING MULTIOBJECTIVE PROGRAM THROUGH ONLINE LEARNING

## ABSTRACT

We investigate the problem of learning the parameters (i.e., objective functions or constraints) of a multiobjective decision making model, based on a set of sequentially arrived decisions. In particular, these decisions might not be exact and possibly carry measurement noise or are generated with the bounded rationality of decision makers. In this paper, we propose a general online learning framework to deal with this learning problem using inverse multiobjective optimization, and prove that this framework converges at a rate of $\mathcal{O}(1/\sqrt{T})$ under certain regularity conditions. More precisely, we develop two online learning algorithms with implicit update rules which can handle noisy data. Numerical results with both synthetic and real world datasets show that both algorithms can learn the parameters of a multiobjective program with great accuracy and are robust to noise.

## 1 INTRODUCTION

In this paper, we aim to learn the parameters (i.e., constraints and a set of objective functions) of a decision making problem with multiple objectives, instead of solving for its efficient (or Pareto) optimal solutions, which is the typical scenario. More precisely, we seek to learn $\theta$ given $\{\mathbf{y}_i\}_{i\in[N]}$ that are observations of the efficient solutions of the multiobjective optimization problem (MOP):

$$\min_{\mathbf{x}} \quad \{f_1(\mathbf{x},\theta), f_2(\mathbf{x},\theta), \dots, f_p(\mathbf{x},\theta)\}$$
$$s.t. \quad \mathbf{x} \in X(\theta),$$

where $\theta$ is the true but unknown parameter of the MOP. In particular, we consider such learning problems in online fashion, noting observations are unveiled sequentially in practical scenarios. Specifically, we study such learning problem as an inverse multiobjective optimization problem (IMOP) dealing with noisy data, develop online learning algorithms to derive parameters for each objective function and constraint, and finally output an estimation of the distribution of weights (which, together with objective functions, define individuals' utility functions) among human subjects.

Learning human participants' decision making scheme is critical for an organization in designing and providing services or products. Nevertheless, as in most scenarios, we can only observe their decisions or behaviors and cannot directly access decision making schemes. Indeed, participants probably do not have exact information regarding their own decision making process (Keshavarz et al., 2011). To bridge the discrepancy, we leverage the inverse optimization idea that has been proposed and received significant attention in the optimization community, which is to infer the missing information of the underlying decision models from observed data, assuming that human decision makers are making optimal decisions (Ahuja & Orlin, 2001; Iyengar & Kang, 2005; Schaefer, 2009; Wang, 2009; Keshavarz et al., 2011; Chan et al., 2014; Bertsimas et al., 2015; Aswani et al., 2018; Esfahani et al., 2018; Tan et al., 2020). This subject actually carries the data-driven concept and becomes more applicable as large amounts of data are generated and become readily available, especially those from digital devices and online transactions.

### 1.1 RELATED WORK

Our work draws inspiration from the inverse optimization problem with single objective. It seeks particular values for those parameters such that the difference between the actual observation and the expected solution to the optimization model (populated with those inferred values) is minimized. Although complicated, an inverse optimization model can often be simplified for computation through

using KKT conditions or strong duality of the decision making model, provided that it is convex. Nowadays, extending from its initial form that only considers a single observation Ahuja & Orlin (2001); Iyengar & Kang (2005); Schaefer (2009); Wang (2009), inverse optimization has been further developed and applied to handle many observations Keshavarz et al. (2011); Bertsimas et al. (2015); Aswani et al. (2018); Esfahani et al. (2018). Nevertheless, a particular challenge, which is almost unavoidable for any large data set, is that the data could be inconsistent due to measurement errors or decision makers' sub-optimality. To address this challenge, the assumption on the observations' optimality is weakened to integrate those noisy data, and KKT conditions or strong duality is relaxed to incorporate inexactness.

Our work is most related to the subject of inverse multiobjective optimization. The goal is to find multiple objective functions or constraints that explain the observed efficient solutions well. There are several recent studies related to the presented research. One is in Chan et al. (2014), which considers a single observation that is assumed to be an exact optimal solution. Then, given a set of well-defined linear functions, an inverse optimization is formulated to learn their weights. Another one is Dong & Zeng (2020), which proposes the batch learning framework to infer utility functions or constraints from multiple noisy decisions through inverse multiobjective optimization. This work can be categorized as doing inverse multiobjective optimization in batch setting. Recently, Dong & Zeng (2021) extends Dong & Zeng (2020) with distributionally robust optimization by leveraging the prominent Wasserstein metric. In contrast, we do inverse multiobjective optimization in online settings, and the proposed online learning algorithms significantly accelerate the learning process with performance guarantees, allowing us to deal with more realistic and complex preference inference problems.

Also related to our work is the line of research conducted by Bärmann et al. (2017) and Dong et al. (2018), which develops online learning methods to infer the utility function or constraints from sequentially arrived observations. However, their approach is only possible to handle inverse optimization with a single objective. More specifically, their methods apply to situations where observations are generated by decision making problems with only one objective function. Differently, our approach does not make the single-objective assumption and only requires the convexity of the underlying decision making problem with multiple objectives. Hence, we believe that our work generalizes their methods and extends the applicability of online learning from learning single objective program to multiobjective program.

## 1.2 OUR CONTRIBUTIONS

To the best of authors' knowledge, we propose the first general framework of online learning for inferring decision makers' objective functions or constraints using inverse multiobjective optimization. This framework can learn the parameters of any convex decision making problem, and can explicitly handle noisy decisions. Moreover, we show that the online learning approach, which adopts an implicit update rule, has an $\mathcal{O}(\sqrt{T})$ regret under suitable regularity conditions when using the ideal loss function. We finally illustrate the performance of two algorithms on both a multiobjective quadratic programming problem and a portfolio optimization problem. Results show that both algorithms can learn parameters with great accuracy and are robust to noise while the second algorithm significantly accelerate the learning process over the first one.

## 2 PROBLEM SETTING

### 2.1 DECISION MAKING PROBLEM WITH MULTIPLE OBJECTIVES

We consider a family of parametrized multiobjective decision making problems of the form

$$\min_{\mathbf{x} \in \mathbb{R}^n} \quad \{f_1(\mathbf{x}, \theta), f_2(\mathbf{x}, \theta), \dots, f_p(\mathbf{x}, \theta)\}$$
$$s.t. \quad \mathbf{x} \in X(\theta), \tag{DMP}$$

where $p \geq 2$ and $f_l(\mathbf{x}, \theta) : \mathbb{R}^n \times \mathbb{R}^{n_\theta} \mapsto \mathbb{R}$ for each $l \in [p]$. Assume parameter $\theta \in \Theta \subseteq \mathbb{R}^{n_\theta}$. We denote the vector of objective functions by $\mathbf{f}(\mathbf{x}, \theta) = (f_1(\mathbf{x}, \theta), f_2(\mathbf{x}, \theta), \dots, f_p(\mathbf{x}, \theta))^T$. Assume $X(\theta) = \{\mathbf{x} \in \mathbb{R}^n : \mathbf{g}(\mathbf{x}, \theta) \leq \mathbf{0}, \mathbf{x} \in \mathbb{R}^n_+\}$, where $\mathbf{g}(\mathbf{x}, \theta) = (g_1(\mathbf{x}, \theta), \dots, g_q(\mathbf{x}, \theta))^T$ is another vector-valued function with $g_k(\mathbf{x}, \theta) : \mathbb{R}^n \times \mathbb{R}^{n_\theta} \mapsto \mathbb{R}$ for each $k \in [q]$.

**Definition 2.1** (Efficiency). For fixed $\theta$, a decision vector $\mathbf{x}^* \in X(\theta)$ is said to be efficient if there exists no other decision vector $\mathbf{x} \in X(\theta)$ such that $f_i(\mathbf{x}, \theta) \leq f_i(\mathbf{x}^*, \theta)$ for all $i \in [p]$, and $f_k(\mathbf{x}, \theta) < f_k(\mathbf{x}^*, \theta)$ for at least one $k \in [p]$.

In the study of multiobjective optimization, the set of all efficient solutions is denoted by $X_E(\theta)$ and called the efficient set. The weighting method is commonly used to obtain an efficient solution through computing the problem of weighted sum (PWS) Gass & Saaty (1955) as follows.

$$\begin{aligned} \min \quad & w^T \mathbf{f}(\mathbf{x}, \theta) \\ s.t. \quad & \mathbf{x} \in X(\theta), \end{aligned} \tag{PWS}$$

where $w = (w^1, \ldots, w^p)^T$. Without loss of generality, all possible weights are restricted to a simplex, which is denoted by $\mathscr{W}_p = \{w \in \mathbb{R}^p_+ : \mathbf{1}^T w = 1\}$. Next, we denote the set of optimal solutions for the (PWS) by

$$S(w, \theta) = \arg\min_{\mathbf{x}} \left\{ w^T \mathbf{f}(\mathbf{x}, \theta) : \mathbf{x} \in X(\theta) \right\}.$$

Let $\mathscr{W}_p^+ = \{w \in \mathbb{R}^p_{++} : \mathbf{1}^T w = 1\}$. Following from Theorem 3.1.2 of Miettinen (2012), we have:

**Proposition 2.1.** If $\mathbf{x} \in S(w, \theta)$ and $w \in \mathscr{W}_p^+$, then $\mathbf{x} \in X_E(\theta)$.

The next result from Theorem 3.1.4 of Miettinen (2012) states that all the efficient solutions can be found by the weighting method for convex MOP.

**Proposition 2.2.** Assume that MOP is convex. If $\mathbf{x} \in X$ is an efficient solution, then there exists a weighting vector $w \in \mathscr{W}_p$ such that $\mathbf{x}$ is an optimal solution of (PWS).

By Propositions 2.1 - 2.2, we can summarize the relationship between $S(w, \theta)$ and $X_E(\theta)$ as follows.

**Corollary 2.2.1.** For convex MOP,

$$\bigcup_{w \in \mathscr{W}_p^+} S(w, \theta) \subseteq X_E(\theta) \subseteq \bigcup_{w \in \mathscr{W}_p} S(w, \theta).$$

In the following, we make a few assumptions to simplify our understanding, which are actually mild and appear often in the literature.

**Assumption 2.1.** Set $\Theta$ is a convex compact set. There exists $D > 0$ such that $\|\theta\|_2 \leq D$ for all $\theta \in \Theta$. In addition, for each $\theta \in \Theta$, both $\mathbf{f}(\mathbf{x}, \theta)$ and $\mathbf{g}(\mathbf{x}, \theta)$ are convex in $\mathbf{x}$.

## 2.2 INVERSE MULTIOBJECTIVE OPTIMIZATION

Consider a learner who has access to decision makers' decisions, but does not know their objective functions or constraints. In our model, the learner aims to learn decision makers' multiple objective functions or constraints from observed noisy decisions only. We denote $\mathbf{y}$ the observed noisy decision that might carry measurement error or is generated with a bounded rationality of the decision maker. We emphasize that this noisy setting of $\mathbf{y}$ reflects the real world situation rather than for analysis of regret. Throughout the paper we assume that $\mathbf{y}$ is a random variable distributed according to an unknown distribution $\mathbb{P}_\mathbf{y}$ supported on $\mathcal{Y}$. As $\mathbf{y}$ is a noisy observation, we note that $\mathbf{y}$ does not necessarily belong to $X(\theta)$, i.e., it might be either feasible or infeasible with respect to $X(\theta)$.

We next discuss the construction of an appropriate loss function for the inverse multiobjective optimization problem Dong & Zeng (2020; 2021). Ideally, given a noisy decision $\mathbf{y}$ and a hypothesis $\theta$, the loss function can be defined as the minimum distance between $\mathbf{y}$ and the efficient set $X_E(\theta)$:

$$l(\mathbf{y}, \theta) = \min_{\mathbf{x} \in X_E(\theta)} \|\mathbf{y} - \mathbf{x}\|_2^2. \tag{loss function}$$

For a general MOP, however, there might exist no explicit way to characterize the efficient set $X_E(\theta)$. Hence, an approximation approach to practically describe this is adopted. Following from Corollary 2.2.1, a sampling approach is adopted to generate $w_k \in \mathscr{W}_p$ for each $k \in [K]$ and approximate $X_E(\theta)$ as $\bigcup_{k \in [K]} S(w_k, \theta)$. Then, the *surrogate loss function* is defined as

$$l_K(\mathbf{y}, \theta) = \min_{\mathbf{x} \in \bigcup_{k \in [K]} S(w_k, \theta)} \|\mathbf{y} - \mathbf{x}\|_2^2. \tag{surrogate loss}$$

By using binary variables, this surrogate loss can be converted into the *Surrogate Loss Problem*.

$$
\begin{aligned}
l_K(\mathbf{y}, \theta) &= \min_{z_j \in \{0,1\}} \|\mathbf{y} - \sum_{k \in [K]} z_k \mathbf{x}_k\|_2^2 \\
&\text{s.t.} \quad \sum_{k \in [K]} z_k = 1, \ \mathbf{x}_k \in S(w_k, \theta).
\end{aligned}
\tag{1}
$$

Constraint $\sum_{k \in [K]} z_k = 1$ ensures that exactly one of the efficient solutions will be chosen to measure the distance to $\mathbf{y}$. Hence, solving this optimization problem identifies some $w_k$ with $k \in [K]$ such that the corresponding efficient solution $S(w_k, \theta)$ is closest to $\mathbf{y}$.

**Remark 2.1.** It is guaranteed that no efficient solution will be excluded if all weight vectors in $\mathscr{W}_p$ are enumerated. As it is practically infeasible due to computational intractability, we can control $K$ to balance the tradeoff between the approximation accuracy and computational efficacy. Certainly, if the computational power is strong, we would suggest to draw a large number of weights evenly in $\mathscr{W}_p$ to avoid any bias. In practice, for general convex MOP, we evenly sample $\{w_k\}_{k \in [K]}$ from $\mathscr{W}_p^+$ to ensure that $S(w_k, \theta) \in X_E(\theta)$. If $\mathbf{f}(\mathbf{x}, \theta)$ is known to be strictly convex, we can evenly sample $\{w_k\}_{k \in [K]}$ from $\mathscr{W}_p$ as $S(w_k, \theta) \in X_E(\theta)$ by Proposition 2.1.

## 3 ONLINE LEARNING FOR IMOP

In our online learning setting, noisy decisions become available to the learner one by one. Hence, the learning algorithm produces a sequence of hypotheses $(\theta_1, \ldots, \theta_{T+1})$. Here, $T$ is the total number of rounds, and $\theta_1$ is an arbitrary initial hypothesis and $\theta_t$ for $t > 1$ is the hypothesis chosen after seeing the $(t-1)$th decision. Let $l(\mathbf{y}_t, \theta_t)$ denote the loss the learning algorithm suffers when it tries to predict $\mathbf{y}_t$ based on the previous observed decisions $\{\mathbf{y}_1, \ldots, \mathbf{y}_{t-1}\}$. The goal of the learner is to minimize the regret, which is the cumulative loss $\sum_{t=1}^{T} l(\mathbf{y}_t, \theta_t)$ against the best possible loss when the whole batch of decisions are available. Formally, the regret is defined as

$$
R_T = \sum_{t=1}^{T} l(\mathbf{y}_t, \theta_t) - \min_{\theta \in \Theta} \sum_{t=1}^{T} l(\mathbf{y}_t, \theta).
$$

Unlike most online learning problems that assume the loss function to be smooth Shalev-Shwartz (2011); Hazan (2016), $l(\mathbf{y}, \theta)$ and $l_K(\mathbf{y}, \theta)$ are not necessarily smooth in our paper, due to the structures of $X_E(\theta)$ and $\bigcup_{k \in [K]} S(w_k, \theta)$. Thus, the popular gradient based online learning algorithms Bottou (1999); Kulis & Bartlett (2010) fail and our problem is significantly more difficult than most of them. To address this challenge, two online learning algorithms are developed in the next section.

### 3.1 ONLINE IMPLICIT UPDATES

Once receiving the $t$th noisy decision $\mathbf{y}_t$, the ideal way to update $\theta_{t+1}$ is by solving the following optimization problem using the ideal loss function:

$$
\theta_{t+1} = \arg\min_{\theta \in \Theta} \frac{1}{2} \|\theta - \theta_t\|_2^2 + \eta_t l(\mathbf{y}_t, \theta),
\tag{2}
$$

where $\eta_t$ is the learning rate in each round, and $l(\mathbf{y}_t, \theta)$ is defined in loss function.

As explained in the previous section, $l(\mathbf{y}_t, \theta)$ might not be computable due to the non-existence of the closed form of the efficient set $X_E(\theta)$. Thus, we seek to approximate the update 2 by:

$$
\theta_{t+1} = \arg\min_{\theta \in \Theta} \frac{1}{2} \|\theta - \theta_t\|_2^2 + \eta_t l_K(\mathbf{y}_t, \theta),
\tag{3}
$$

where $\eta_t$ is the learning rate in each round, and $l_K(\mathbf{y}_t, \theta)$ is defined in surrogate loss.

The update 3 approximates 2, and seeks to balance the tradeoff between "conservativeness" and "correctiveness", where the first term characterizes how conservative we are to maintain the current estimation, and the second term indicates how corrective we would like to modify with the new estimation. As no closed form exists for $\theta_{t+1}$ in general, this update method is an implicit approach.

| **Algorithm 1** Online Learning for IMOP | **Algorithm 2** Accelerated Online Learning |
|---|---|
| 1: **Input:** noisy decisions $\{\mathbf{y}_t\}_{t \in T}$, weights $\{w_k\}_{k \in K}$ | 1: **Input:** $\{\mathbf{y}_t\}_{t \in T}$ and $\{w_k\}_{k \in K}$ |
| 2: **Initialize** $\theta_1 = \mathbf{0}$ | 2: **Initialize** $\theta_1 = \mathbf{0}$ |
| 3: **for** $t = 1$ to $T$ **do** | 3: **for** $t = 1$ to $T$ **do** |
| 4:    receive $\mathbf{y}_t$ | 4:    receive $\mathbf{y}_t$ |
| 5:    suffer loss $l_K(\mathbf{y}_t, \theta_t)$ | 5:    suffer loss $l_K(\mathbf{y}_t, \theta_t)$ |
| 6:   **if** $l_K(\mathbf{y}_t, \theta_t) = 0$ **then** | 6:    let $k^* = \arg\min_{k \in [K]} \|\mathbf{y}_t - \mathbf{x}_k\|_2^2$, where $\mathbf{x}_k \in S(w_k, \theta_t)$ for $k \in [K]$ |
| 7:      $\theta_{t+1} \leftarrow \theta_t$ | 7:   **if** $l_K(\mathbf{y}_t, \theta_t) = 0$ **then** |
| 8:   **else** | 8:      $\theta_{t+1} \leftarrow \theta_t$ |
| 9:      set learning rate $\eta_t \propto 1/\sqrt{t}$ | 9:   **else** |
| 10:     update $\theta_{t+1}$ by solving 3 directly (or equivalently solving $K$ subproblems 4) | 10:     set learning rate $\eta_t \propto 1/\sqrt{t}$ |
| 11:   **end if** | 11:     update $\theta_{t+1}$ by 4 with $k = k^*$ |
| 12: **end for** | 12:   **end if** |
|  | 13: **end for** |

To solve 3, we can replace $\mathbf{x}_k \in S(w_k, \theta)$ by KKT conditions for each $k \in [K]$:

$$
\begin{aligned}
\min_{\theta} \quad & \tfrac{1}{2}\|\theta - \theta_t\|_2^2 + \eta_t \sum_{k \in [K]} \|\mathbf{y}_t - \vartheta_k\|_2^2 \\
\text{s.t.} \quad & \theta \in \Theta, \\
& \left[\begin{array}{l} \mathbf{g}(\mathbf{x}_k) \leq \mathbf{0}, \ \mathbf{u}_k \geq \mathbf{0}, \\ \mathbf{u}_k^T \mathbf{g}(\mathbf{x}_k) = 0, \\ \nabla_{\mathbf{x}_k} w_k^T \mathbf{f}(\mathbf{x}_k, \theta) + \mathbf{u}_k \cdot \nabla_{\mathbf{x}_k}\mathbf{g}(\mathbf{x}_k) = \mathbf{0}, \end{array}\right], \quad \forall k \in [K], \\
& 0 \leq \vartheta_k \leq M_k z_k, && \forall k \in [K], \\
& \mathbf{x}_k - M_k(1 - z_k) \leq \vartheta_k \leq \mathbf{x}_k, && \forall k \in [K], \\
& \sum_{k \in [K]} z_k = 1, \\
& \mathbf{x}_k \in \mathbb{R}^n, \ \mathbf{u}_k \in \mathbb{R}_+^m, \ z_k \in \{0, 1\}, && \forall k \in [K],
\end{aligned}
$$

where $\mathbf{u}_k$ is the dual variable for $g_k(\mathbf{x}, \theta) \leq 0$, and $M_k$ is a big number to linearize $z_k \mathbf{x}_k$.

Alternatively, solving 3 is equivalent to solving $K$ independent programs defined in the following and taking the one with the least optimal value (breaking ties arbitrarily).

$$
\begin{aligned}
\min_{\theta \in \Theta} \quad & \tfrac{1}{2}\|\theta - \theta_t\|_2^2 + \eta_t \|\mathbf{y}_t - \mathbf{x}\|_2^2 \\
\text{s.t.} \quad & \mathbf{x} \in S(w_k, \theta).
\end{aligned}
\tag{4}
$$

Our application of the implicit update rule to learn an MOP proceeds as outlined in Algorithm 1.

**Remark 3.1.** $(i)$ When choosing 4 to update $\theta_{t+1}$, we can parallelly compute $K$ independent problems of 4, which would dramatically improve the computational efficiency. $(ii)$ After the completion of Algorithm 1, we can allocate every $\mathbf{y}_t$ to the $w_k$ that minimizes $l_K(\mathbf{y}_t, \theta_{T+1})$, which provides an inference on the distribution of weights of component functions $f_l(\mathbf{x}, \theta)$ over human subjects.

**Acceleration of Algorithm** 1: Note that we update $\theta$ and the weight sample assigned to $\mathbf{y}_t$ in 3 simultaneously, meaning both $\theta$ and the weight sample index $k$ are variables when solving 3. In other words, one needs to solve $K$ subproblems 4 to get an optimal solution for 3. However, note that the increment of $\theta$ by 3 is typically small for each update. Consequently, the weight sample assigned to $\mathbf{y}_t$ using $\theta_{t+1}$ is roughly the same as using the previous guess of this parameter, i.e., $\theta_t$. Hence, it is reasonable to approximate 3 by first assigning a weight sample to $\mathbf{y}_t$ based on the previous updating result. Then, instead of computing $K$ problems of 4, we simply compute a single one associated with the selected weight samples, which significantly eases the burden of solving 3. Our application of the accelerated implicit update rule proceeds as outlined in Algorithm 2.

**Mini-batches** We enhance online learning by considering multiple observations per update Bottou & Cun (2004). In online IMOP, this means that computing $\theta_{t+1}$ using $|N_t| > 1$ decisions:

$$\theta_{t+1} = \arg\min_{\theta \in \Theta} \frac{1}{2}\|\theta - \theta_t\|_2^2 + \frac{\eta_t}{|N_t|} \sum_{t \in N_t} l_K(\mathbf{y}_t, \theta), \tag{5}$$

However, we should point out that applying Mini-batches might not be suitable here as the update 5 is drastically more difficult to compute even for $|N_t| = 2$ than the update 3 with a single observation.

## 3.2 ANALYSIS OF CONVERGENCE

Note that the proposed online learning algorithms are generally applicable to learn the parameter of any convex MOP. In this section, we show that the average regret converges at a rate of $\mathcal{O}(1/\sqrt{T})$ under certain regularity conditions based on the ideal loss function $l(\mathbf{y}, \theta)$. Namely, we consider the regret bound when using the ideally implicit update rule 2. Next, we introduce a few assumptions that are regular in literature Keshavarz et al. (2011); Bertsimas et al. (2015); Esfahani et al. (2018); Aswani et al. (2018); Dong & Zeng (2018); Dong et al. (2018).

**Assumption 3.1. (a)** $X(\theta)$ is closed, and has a nonempty relative interior. $X(\theta)$ is also bounded. Namely, there exists $B > 0$ such that $\|\mathbf{x}\|_2 \leq B$ for all $\mathbf{x} \in X(\theta)$. The support $\mathcal{Y}$ of the noisy decisions $\mathbf{y}$ is contained within a ball of radius $R$ almost surely, where $R < \infty$. In other words, $\mathbb{P}(\|\mathbf{y}\|_2 \leq R) = 1$.

**(b)** Each function in $\mathbf{f}$ is strongly convex on $\mathbb{R}^n$, that is for each $l \in [p], \exists \lambda_l > 0, \forall \mathbf{x}, \mathbf{y} \in \mathbb{R}^n$

$$\left(\nabla f_l(\mathbf{y}, \theta_l) - \nabla f_l(\mathbf{x}, \theta_l)\right)^T (\mathbf{y} - \mathbf{x}) \geq \lambda_l \|\mathbf{x} - \mathbf{y}\|_2^2.$$

Regarding Assumption 3.1.(a), assuming that the feasible region is closed and bounded is very common in inverse optimization. The finite support of the observations is needed since we do not hope outliers have too many impacts in our learning. Let $\lambda = \min_{l \in [p]}\{\lambda_l\}$. It follows that $w^T \mathbf{f}(\mathbf{x}, \theta)$ is strongly convex with parameter $\lambda$ for $w \in \mathscr{W}_p$. Therefore, Assumption 3.1.(b) ensures that $S(w, \theta)$ is a single-valued set for each $w$.

The performance of the algorithm also depends on how the change of $\theta$ affects the objective values. For $\forall w \in \mathscr{W}_p, \theta_1 \in \Theta, \theta_2 \in \Theta$, we consider the following function

$$h(\mathbf{x}, w, \theta_1, \theta_2) = w^T \mathbf{f}(\mathbf{x}, \theta_1) - w^T \mathbf{f}(\mathbf{x}, \theta_2).$$

**Assumption 3.2.** $\exists \kappa > 0, \forall w \in \mathscr{W}_p, h(\cdot, w, \theta_1, \theta_2)$ is $\kappa$-Lipschitz continuous on $\mathcal{Y}$. That is,

$$|h(\mathbf{x}, w, \theta_1, \theta_2) - h(\mathbf{y}, w, \theta_1, \theta_2)| \leq \kappa \|\theta_1 - \theta_2\|_2 \|\mathbf{x} - \mathbf{y}\|_2, \forall \mathbf{x}, \mathbf{y} \in \mathcal{Y}.$$

Basically, this assumption says that the objective functions will not change much when either the parameter $\theta$ or the variable $\mathbf{x}$ is perturbed. It actually holds in many common situations, including the multiobjective linear program and multiobjective quadratic program.

From now on, given any $\mathbf{y} \in \mathcal{Y}, \theta \in \Theta$, we denote $\mathbf{x}(\theta)$ the efficient point in $X_E(\theta)$ that is closest to $\mathbf{y}$. Namely, $l(\mathbf{y}, \theta) = \|\mathbf{y} - \mathbf{x}(\theta)\|_2^2$.

**Lemma 3.1.** Under Assumptions 3.1 - 3.2, the loss function $l(\mathbf{y}, \theta)$ is uniformly $\frac{4(B+R)\kappa}{\lambda}$-Lipschitz continuous in $\theta$. That is, $\forall \mathbf{y} \in \mathcal{Y}, \forall \theta_1, \theta_2 \in \Theta$, we have

$$|l(\mathbf{y}, \theta_1) - l(\mathbf{y}, \theta_2)| \leq \frac{4(B+R)\kappa}{\lambda}\|\theta_1 - \theta_2\|_2.$$

The key point in proving Lemma 3.1 is the observation that the perturbation of $S(w, \theta)$ due to $\theta$ is bounded by the perturbation of $\theta$ by applying Proposition 6.1 in Bonnans & Shapiro (1998). Details of the proof are given in Appendix.

**Assumption 3.3.** For MOP, $\forall \mathbf{y} \in \mathcal{Y}, \forall \theta_1, \theta_2 \in \Theta, \forall \alpha, \beta \geq 0$ s.t. $\alpha + \beta = 1$, we have either of the following:

**(a)** if $\mathbf{x}_1 \in X_E(\theta_1)$, and $\mathbf{x}_2 \in X_E(\theta_2)$, then $\alpha \mathbf{x}_1 + \beta \mathbf{x}_2 \in X_E(\alpha \theta_1 + \beta \theta_2)$.

**(b)** $\|\alpha\mathbf{x}(\theta_1) + \beta\mathbf{x}(\theta_2) - \mathbf{x}(\alpha\theta_1 + \beta\theta_2)\|_2 \leq \alpha\beta\|\mathbf{x}(\theta_1) - \mathbf{x}(\theta_2)\|_2/(2(B+R))$.

The definition of $\mathbf{x}(\theta_1), \mathbf{x}(\theta_2)$ and $\mathbf{x}(\alpha\theta_1 + \beta\theta_2)$ is given before Lemma 3.1. This assumption requires the convex combination of $\mathbf{x}_1 \in X_E(\theta_1)$, and $\mathbf{x}_2 \in X_E(\theta_2)$ belongs to $X_E(\alpha\theta_1 + \beta\theta_2)$. Or there exists an efficient point in $X_E(\alpha\theta_1 + \beta\theta_2)$ close to the convex combination of $\mathbf{x}(\theta_1)$ and $\mathbf{x}(\theta_2)$. Examples are given in Appendix.

Let $\theta^*$ be an optimal inference to $\min_{\theta\in\Theta} \sum_{t\in[T]} l(\mathbf{y}_t, \theta)$, i.e., an inference derived with the whole batch of observations available. Then, the following theorem asserts that under the above assumptions, the regret $R_T = \sum_{t\in[T]}(l(\mathbf{y}_t, \theta_t) - l(\mathbf{y}_t, \theta^*))$ of the online learning algorithm is of $\mathcal{O}(\sqrt{T})$.

**Theorem 3.2.** Suppose Assumptions 3.1 - 3.3 hold. Then, choosing $\eta_t = \frac{D\lambda}{2\sqrt{2}(B+R)\kappa}\frac{1}{\sqrt{t}}$, we have

$$R_T \leq \frac{4\sqrt{2}(B+R)D\kappa}{\lambda}\sqrt{T}.$$

We establish the above regret bound by extending Theorem 3.2 in Kulis & Bartlett (2010). Our extension involves several critical and complicated analyses for the structure of the optimal solution set $S(w, \theta)$ as well as the loss function, which is essential to our theoretical understanding. Moreover, we relax the requirement of smoothness of loss function to Lipschitz continuity through a similar argument in Lemma 1 of Wang et al. (2017) and Duchi et al. (2011).

## 4 EXPERIMENTS

In this section, we will provide a multiobjective quadratic program (MQP) and a portfolio optimization problem to illustrate the performance of the proposed online learning Algorithms 1 and 2. The mixed integer second order conic problems (MISOCPs), which are derived from using KKT conditions in 3, are solved by Gurobi Optimization (2016). All the algorithms are programmed with Julia Bezanson et al. (2017). The experiments have been run on an Intel(R) Xeon(R) E5-1620 processor that has a 3.60GHz CPU with 32 GB RAM.

### 4.1 SYNTHETIC DATA: LEARNING THE PREFERENCES AND RESTRICTIONS FOR AN MQP

Consider the following multiobjective quadratic optimization problem.

$$\min_{\mathbf{x}\in\mathbb{R}^2_+} \quad \begin{pmatrix} f_1(\mathbf{x}) = \frac{1}{2}\mathbf{x}^T Q_1\mathbf{x} + \mathbf{c}_1^T\mathbf{x} \\ f_2(\mathbf{x}) = \frac{1}{2}\mathbf{x}^T Q_2\mathbf{x} + \mathbf{c}_2^T\mathbf{x} \end{pmatrix}$$
$$s.t. \quad A\mathbf{x} \leq \mathbf{b},$$

where parameters of the objective functions and constraints are provided in Appendix.

Suppose there are $T$ decision makers. In each round, the learner would receive one noisy decision. Her goal is to learn the objective functions or restrictions of these decision makers. In round $t$, we suppose that the decision maker derives an efficient solution $\mathbf{x}_t$ by solving (PWS) with weight $w_t$, which is uniformly chosen from $\mathscr{W}_2$. Next, the learner receives the noisy decision $\mathbf{y}_t$ corrupted by noise that has a jointly uniform distribution with support $[-0.5, 0.5]^2$. Namely, $\mathbf{y}_t = \mathbf{x}_t + \epsilon_t$, where each element of $\epsilon_t \sim U(-0.5, 0.5)$.

**Learning the objective functions** In the first set of experiments, the learner seeks to learn $\mathbf{c}_1$ and $\mathbf{c}_2$ given the noisy decisions that arrive sequentially in $T$ rounds. We assume that $\mathbf{c}_1$ is within range $[1, 6]^2$, $\mathbf{c}_2$ is within range $[-6, -1]^2$, $T = 1000$ rounds of noisy decisions are generated, and $K = 41$ weights from $\mathscr{W}_2$ are evenly sampled. The learning rate is set to $\eta_t = 5/\sqrt{t}$. Then, we implement Algorithms 1 and 2. At each round $t$, we solve 4 using parallel computing with 6 workers.

To illustrate the performance of the algorithms in a statistical way, we run 100 repetitions of the experiments. Figure 1a shows the total estimation errors of $\mathbf{c}_1$ and $\mathbf{c}_2$ in each round over the 100 repetitions for the two algorithms. We also plot the average estimation error of the 100 repetitions. As can be seen in this figure, convergence for both algorithms is pretty fast. Also, estimation errors over rounds for different repetitions concentrate around the average, indicating that our algorithm is pretty robust to noise. The estimation error in the last round is not zero because we use a finite $K$ to approximate the efficient set. We see in Figure 1b that Algorithm 2 is much faster than Algorithm 1 especially when $K$ is large. To further illustrate the performance of algorithms, we randomly pick one repetition using Algorithm 1 and plot the estimated efficient set in Figure 1c. We can see

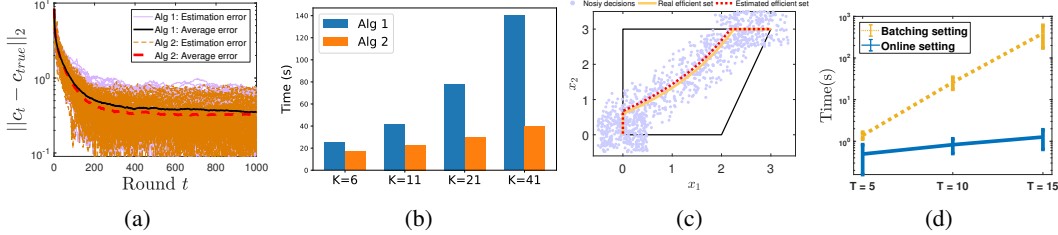

Figure 1: Learning objective functions of an MQP over $T = 1000$ rounds. We run 100 repetitions of experiments. Let $\mathbf{c} = [\mathbf{c}_1, \mathbf{c}_2]$. (a) We plot estimation errors at each round $t$ for all 100 experiments and their average estimation errors with $K = 41$. (b) Blue and yellow bars indicate average running time and standard deviations for each $K$ using Algorithm 1 and 2, respectively. (c) We randomly pick one repetition. The estimated efficient set after $T = 1000$ rounds is indicated by the red line. The real efficient set is shown by the yellow line. (d) The dotted brown line is the error bar plot of the running time over 10 repetitions in batch setting. The blue line is the error bar plot of the running time over 100 repetitions in an online setting using Algorithm 1.

clearly that the estimated efficient set almost coincides with the real efficient set. Moreover, Figure 1d shows that IMOP in online settings is drastically faster than in batch setting. It is practically impossible to apply the batch setting algorithms in real-world applications.

**Learning the Right-hand Side** In the second set of experiments, the learner seeks to learn $\mathbf{b}$ given the noisy decisions that arrive sequentially in $T$ rounds. We assume that $\mathbf{b}$ is within $[-10, 10]^2$. $T = 1000$ rounds of noisy decisions are generated. $K = 81$ weights from $\mathscr{W}_2$ are evenly sampled. The learning rate is set to $\eta_t = 5/\sqrt{t}$. Then, we apply Algorithms 1 and 2. To illustrate the performance of them, we run 100 repetitions of the experiments. Figure 2a shows the estimation error of $\mathbf{b}$ in each round over the 100 repetitions for the two algorithms. We also plot the average estimation error of the 100 repetitions. As can be seen in the figure, convergence for both algorithms is pretty fast. In addition, we see in Figure 2b that Algorithm 2 is much faster than Algorithm 1.

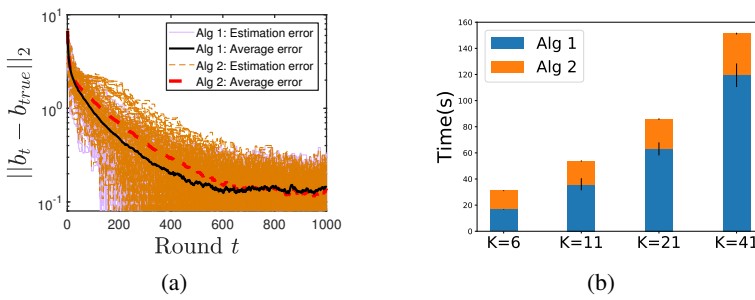

Figure 2: Learning the right-hand side of an MQP over $T = 1000$ rounds. We run 100 repetitions of the experiments. (a) We plot estimation errors at each round $t$ for all 100 experiments and their average estimation errors of all repetitions with $K = 41$. (b) Blue and yellow bars indicate the average running times and standard deviations for each $K$ using Algorithm 1 and 2, respectively.

## 4.2 REAL-WORLD CASE: LEARNING EXPECTED RETURNS IN PORTFOLIO OPTIMIZATION

We next consider noisy decisions arising from different investors in a stock market. More precisely, we consider a portfolio selection problem, where investors need to determine the fraction of their wealth to invest in each security to maximize the total return and minimize the total risk. The process typically involves the cooperation between an investor and a portfolio analyst, where the analyst provides an efficient frontier on a certain set of securities to the investor and then the investor selects a portfolio according to her preference to the returns and risks. The classical Markovitz mean-variance portfolio selection Markowitz (1952) in the following is used by analysts.

$$\min \quad \begin{pmatrix} f_1(\mathbf{x}) & = -\mathbf{r}^T\mathbf{x} \\ f_2(\mathbf{x}) & = \mathbf{x}^T Q \mathbf{x} \end{pmatrix}$$
$$s.t. \quad 0 \le x_i \le b_i, \qquad \forall i \in [n],$$
$$\sum_{i=1}^{n} x_i = 1,$$

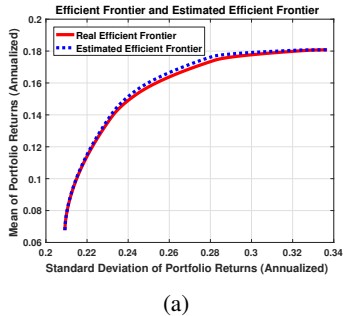
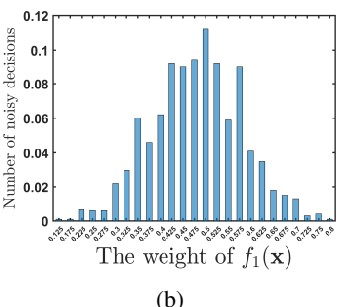

(a)                          (b)

Figure 3: Learning the expected return of a Portfolio optimization problem over $T = 1000$ rounds with $K = 41$. (a) The red line indicates the real efficient frontier. The blue dots indicate the estimated efficient frontier using the estimated expected return for $K = 41$. (b) Each bar represents the proportion of the 1000 decision makers that has the corresponding weight for $f_1(\mathbf{x})$.

where $\mathbf{r} \in \mathbb{R}_+^n$ is a vector of individual security expected returns, $Q \in \mathbb{R}^{n \times n}$ is the covariance matrix of securities returns, $\mathbf{x}$ is a portfolio specifying the proportions of capital to be invested in the different securities, and $b_i$ is an upper bound on the proportion of security $i$, $\forall i \in [n]$.

**Dataset**: The dataset is derived from monthly total returns of 30 stocks from a blue-chip index which tracks the performance of top 30 stocks in the market when the total investment universe consists of thousands of assets. The true expected returns and true return covariance matrix for the first 8 securities are given in the Appendix.

Details for generating the portfolios are provided in Appendix. The portfolios on the efficient frontier are plot in Figure 3a. The learning rate is set to $\eta_t = 5/\sqrt{t}$. At each round $t$, we solve 4 using parallel computing. In Table 1 we list the estimation error and estimated expected returns for different $K$. The estimation error becomes smaller when $K$ increases, indicating that we have a better approximation accuracy of the efficient set when using a larger $K$. We also plot the estimated efficient frontier using the estimated $\hat{\mathbf{r}}$ for $K = 41$ in Figure 3a. We can see that the estimated efficient frontier is very close to the real one, showing that our algorithm works quite well in learning expected returns in portfolio optimization. We also plot our estimation on the distribution of the weight of $f_1(\mathbf{x})$ among the 1000 decision makers. As shown in Figure 3b, the distribution follows roughly normal distribution. We apply Chi-square goodness-of-fit tests to support our hypotheses.

Table 1: Estimation Error for Different $K$

| $K$ | 6 | 11 | 21 | 41 |
|---|---|---|---|---|
| $\|\hat{\mathbf{r}} - \mathbf{r}_{true}\|_2$ | 0.1270 | 0.1270 | 0.0420 | 0.0091 |

## 5   CONCLUSION AND FUTURE WORK

In this paper, an online learning method to learn the parameters of the multiobjective optimization problems from noisy observations is developed and implemented. We prove that this framework converges at a rate of $\mathcal{O}(1/\sqrt{T})$ under suitable conditions. Nonetheless, as shown in multiple experiments using both synthetic and real world datasets, even if these conditions are not satisfied, we still observe a fast convergence rate and a strong robustness to noisy observations. Thus, it would be interesting to analyze to what extent these conditions can be relaxed.

Also, we note that our model naturally follows the contextual bandit setting. We can view the decision $y_t$ observed at time $t$ as the context. The learner then takes the action $\theta_t$ and the loss is jointly determined by the context and the action. Compared to the vast majority of literature surveyed in Zhou (2015), the main technical difficulty in our model is how to set an appropriate reward (loss) given the context ($y_t$) and the action ($\theta_t$). Intuitively, we set the loss as the difference between the context ($y_t$) and another context generated by the action. Motivated by this observation, one future work is to integrate classical contextual bandits algorithms into our model. Particularly, we think that algorithms without the Linear Realizability Assumption (the reward is linear with respect to the context), such as KernelUCB, might fit well in our problem.

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

## A  APPENDIX

### A.1  OMITTED MATHEMATICAL REFORMULATIONS

Before giving the reformulations, we first make some discussions about the surrogate loss functions.

$$l_K(\mathbf{y}, \theta) = \min_{z_k \in \{0,1\}} \|\mathbf{y} - \sum_{k \in [K]} z_k \mathbf{x}_k\|_2^2$$

$$= \min_{z_k \in \{0,1\}} \sum_{k \in [K]} \|\mathbf{y} - z_k \mathbf{x}_k\|_2^2 - (K-1)\|\mathbf{y}\|_2^2$$

where $\mathbf{x}_k \in S(w_k, \theta)$ and $\sum_{k \in [K]} z_k = 1$.

Since $(K-1)\|\mathbf{y}\|_2^2$ is a constant, we can safely drop it and use the following as the surrogate loss function when solving the optimization program in the implicit update,

$$l_K(\mathbf{y}, \theta) = \min_{z_k \in \{0,1\}} \sum_{k \in [K]} \|\mathbf{y} - z_k \mathbf{x}_k\|_2^2$$

where $\mathbf{x}_k \in S(w_k, \theta)$ and $\sum_{k \in [K]} z_k = 1$.

#### A.1.1  SINGLE LEVEL REFORMULATION FOR THE INVERSE MULTIOBJECTIVE OPTIMIZATION PROBLEM

The parametrized mulobjective optimization problem is

$$\begin{aligned} \min_{\mathbf{x} \in \mathbb{R}^n} \quad & \mathbf{f}(\mathbf{x}, \theta) \\ s.t. \quad & \mathbf{g}(\mathbf{x}) \leq \mathbf{0} \end{aligned} \qquad \text{MOP}$$

where

$$\mathbf{f}(\mathbf{x}, \theta) = (f_1(\mathbf{x}, \theta), f_2(\mathbf{x}, \theta), \ldots, f_p(\mathbf{x}, \theta))^T$$
$$\mathbf{g}(\mathbf{x}) = (g_1(\mathbf{x}), \ldots, g_q(\mathbf{x}))^T$$

Then, the single level reformulation for the Implicit update in the paper is given in the following

$$
\begin{aligned}
\min_{\mathbf{b}} \quad & \tfrac{1}{2}\|\theta - \theta_t\|_2^2 + \eta_t \sum_{k\in[K]} \|\mathbf{y}_t - \vartheta_k\|_2^2 \\
\text{s.t.} \quad & \theta \in \Theta \\
& \left[
\begin{array}{l}
\mathbf{g}(\mathbf{x}_k) \leq \mathbf{0}, \ \ \mathbf{u}_k \geq \mathbf{0} \\
\mathbf{u}_k^T \mathbf{g}(\mathbf{x}_k) = 0 \\
\nabla_{\mathbf{x}_k} w_k^T \mathbf{f}(\mathbf{x}_k, \theta) + \mathbf{u}_k \cdot \nabla_{\mathbf{x}_k} \mathbf{g}(\mathbf{x}_k) = \mathbf{0}
\end{array}
\right] \quad && \forall k \in [K] \\
& 0 \leq \vartheta_k \leq M_k z_k && \forall k \in [K] \\
& \mathbf{x}_k - M_k(1 - z_k) \leq \vartheta_k \leq \mathbf{x}_k && \forall k \in [K] \\
& \sum_{k\in[K]} z_k = 1 \\
& \mathbf{x}_k \in \mathbb{R}^n, \ \ \mathbf{u}_k \in \mathbb{R}_+^m, \ \ \mathbf{t}_k \in \{0,1\}^m, \ \ z_k \in \{0,1\} && \forall k \in [K]
\end{aligned}
$$

### A.1.2 SINGLE LEVEL REFORMULATION FOR THE INVERSE MULTIOBJECTIVE QUADRATIC PROBLEM

When the objective functions are quadratic and the feasible region is a polyhedron, the multiobjective optimization has the following form

$$
\min_{\mathbf{x} \in \mathbb{R}^n} \quad
\begin{bmatrix}
\tfrac{1}{2}\mathbf{x}^T Q_1 \mathbf{x} + \mathbf{c}_1^T \mathbf{x} \\
\vdots \\
\tfrac{1}{2}\mathbf{x}^T Q_p \mathbf{x} + \mathbf{c}_p^T \mathbf{x}
\end{bmatrix}
\qquad\qquad \text{MQP}
$$
$$
s.t. \quad A\mathbf{x} \geq \mathbf{b}
$$

where $Q_l \in \mathbf{S}_+^n$ (the set of symmetric positive semidefinite matrices) for all $l \in [p]$..

When trying to learn $\{\mathbf{c}_l\}_{l\in[p]}$, the single level reformulation for the Implicit update in the paper is given in the following

$$
\begin{aligned}
\min_{\mathbf{c}_l} \quad & \tfrac{1}{2}\sum_{l\in[p]} \|\mathbf{c}_l - \mathbf{c}_l^t\|_2^2 + \eta_t \sum_{k\in[K]} \|\mathbf{y}_t - \vartheta_k\|_2^2 \\
\text{s.t.} \quad & \mathbf{c}_l \in \widetilde{C}_l && \forall l \in [p] \\
& \left[
\begin{array}{l}
\mathbf{A}\mathbf{x}_k \geq \mathbf{b}, \ \mathbf{u}_k \geq \mathbf{0} \\
\mathbf{u}_k \leq M \mathbf{t}_k \\
\mathbf{A}\mathbf{x}_k - \mathbf{b} \leq M(1 - \mathbf{t}_k) \\
(w_k^1 Q_1 + \cdots + w_k^p Q_p)\mathbf{x}_i + w_k^1 \mathbf{c}_1 + \cdots + w_k^p \mathbf{c}_p - \mathbf{A}^T \mathbf{u}_k = 0
\end{array}
\right] \quad && \forall k \in [K] \\
& 0 \leq \vartheta_k \leq M_k z_k && \forall k \in [K] \\
& \mathbf{x}_k - M_k(1 - z_k) \leq \vartheta_k \leq \mathbf{x}_k && \forall k \in [K] \\
& \sum_{k\in[K]} z_k = 1 \\
& \mathbf{x}_k \in \mathbb{R}^n, \ \ \mathbf{u}_k \in \mathbb{R}_+^m, \ \ \mathbf{t}_k \in \{0,1\}^m, \ \ z_k \in \{0,1\} && \forall l \in [p] \ \ \forall k \in [K]
\end{aligned}
$$

where $\mathbf{c}_l^t$ is the estimation of $\mathbf{c}_l$ at the $t$th round, and $\widetilde{C}_l$ is a convex set for each $l \in [p]$.

We have a similar single level reformulation when learning the Right-hand side $\mathbf{b}$. Clearly, this is a Mixed Integer Second Order Cone program(MISOCP) when learning either $\mathbf{c}_l$ or $\mathbf{b}$.

## A.2  OMITTED PROOFS

### A.2.1  STRONGLY CONVEX OF $w^T \mathbf{f}(\mathbf{x}, \theta)$ AS STATED UNDER ASSUMPTION 3.1

*Proof.* By the definition of $\lambda$,

$$
\begin{aligned}
\left( \nabla w^T \mathbf{f}(\mathbf{y}, \theta) - \nabla w^T \mathbf{f}(\mathbf{x}, \theta) \right)^T (\mathbf{y} - \mathbf{x}) &= \left( \nabla \sum_{l=1}^{p} w_l f_l(\mathbf{y}, \theta) - \nabla \sum_{l=1}^{p} w_l f_l(\mathbf{x}, \theta_l) \right)^T (\mathbf{y} - \mathbf{x}) \\
&= \sum_{l=1}^{p} w_l \left( \nabla f_l(\mathbf{y}, \theta_l) - \nabla f_l(\mathbf{x}, \theta_l) \right)^T (\mathbf{y} - \mathbf{x}) \\
&\geq \sum_{l=1}^{p} w_l \lambda_l \|\mathbf{x} - \mathbf{y}\|_2^2 \geq \eta \|\mathbf{x} - \mathbf{y}\|_2^2 \sum_{l=1}^{p} w_l \\
&= \lambda \|\mathbf{x} - \mathbf{y}\|_2^2
\end{aligned}
$$

Thus, $w^T \mathbf{f}(\mathbf{x}, \theta)$ is strongly convex for $\mathbf{x} \in \mathbb{R}^n$. $\qquad \square$

### A.2.2  PROOF OF LEMMA 3.1

*Proof.* By Assumption 3.1(b), we know that $S(w, \theta)$ is a single-valued set for each $w \in \mathscr{W}_p$. Thus, $\forall \mathbf{y} \in \mathcal{Y}, \forall \theta_1, \theta_2 \in \Theta, \exists w^1, w^2 \in \mathscr{W}_p$, s.t.

$$
\mathbf{x}(\theta_1) = S(w^1, \theta_1), \quad \mathbf{x}(\theta_2) = S(w^2, \theta_2)
$$

Without of loss of generality, let $l_K(\mathbf{y}, \theta_1) \geq l_K(\mathbf{y}, \theta_2)$. Then,

$$
\begin{aligned}
|l_K(\mathbf{y}, \theta_1) - l_K(\mathbf{y}, \theta_2)| &= l_K(\mathbf{y}, \theta_1) - l_K(\mathbf{y}, \theta_2) \\
&= \|\mathbf{y} - \mathbf{x}(\theta_1)\|_2^2 - \|\mathbf{y} - \mathbf{x}(\theta_2)\|_2^2 \\
&= \|\mathbf{y} - S(w^1, \theta_1)\|_2^2 - \|\mathbf{y} - S(w^2, \theta_2)\|_2^2 \\
&\leq \|\mathbf{y} - S(w^2, \theta_1)\|_2^2 - \|\mathbf{y} - S(w^2, \theta_2)\|_2^2 \\
&= \langle S(w^2, \theta_2) - S(w^2, \theta_1), 2\mathbf{y} - S(w^2, \theta_1) - S(w^2, \theta_2) \rangle \\
&\leq 2(B + R)\|S(w^2, \theta_2) - S(w^2, \theta_1)\|_2
\end{aligned} \tag{6}
$$

The last inequality is due to Cauchy-Schwartz inequality and the Assumptions 3.1(a), that is

$$
\|2\mathbf{y} - S(w^2, \theta_1) - S(w^2, \theta_2)\|_2 \leq 2(B + R) \tag{7}
$$

Next, we will apply Proposition 6.1 in Bonnans & Shapiro (1998) to bound $\|S(w^2, \theta_2) - S(w^2, \theta_1)\|_2$.

Under Assumptions 3.1 - 3.2, the conditions of Proposition 6.1 in Bonnans & Shapiro (1998) are satisfied. Therefore,

$$
\|S(w^2, \theta_2) - S(w^2, \theta_1)\|_2 \leq \frac{2\kappa}{\lambda} \|\theta_1 - \theta_2\|_2 \tag{8}
$$

Plugging equation 7 and equation 8 in equation 6 yields the claim. $\qquad \square$

### A.2.3  PROOF OF THEOREM 3.2

*Proof.* We will extend Theorem 3.2 in Kulis & Bartlett (2010) to prove our theorem.

Let $G_t(\theta) = \frac{1}{2}\|\theta - \theta_t\|_2^2 + \eta_t l(\mathbf{y}_t, \theta)$.

We will now show the loss function is convex. The first step is to show that if Assumption 3.3 holds, then the loss function $l(\mathbf{y}, \theta)$ is convex in $\theta$.

First, suppose Assumption 3.3(a) hold. Then,

$$
\begin{aligned}
& \alpha l(\mathbf{y}, \theta_1) + \beta l(\mathbf{y}, \theta_2) - l(\mathbf{y}, \alpha\theta_1 + \beta\theta_2) \\
=\ & \alpha\|\mathbf{y} - \mathbf{x}(\theta_1)\|_2^2 + \beta\|\mathbf{y} - \mathbf{x}(\theta_2)\|_2^2 - \|\mathbf{y} - \mathbf{x}(\alpha\theta_1 + \beta\theta_2)\|_2^2 \\
\geq\ & \alpha\|\mathbf{y} - \mathbf{x}(\theta_1)\|_2^2 + \beta\|\mathbf{y} - \mathbf{x}(\theta_2)\|_2^2 - \|\mathbf{y} - \alpha\mathbf{x}(\theta_1) - \beta\mathbf{x}(\theta_2)\|_2^2 \quad \text{(By Assumption 3.3(a))} \\
=\ & \alpha\beta\|\mathbf{x}(\theta_1) - \mathbf{x}(\theta_2)\|_2^2 \\
\geq\ & 0
\end{aligned}
\tag{9}
$$

Second, suppose Assumption 3.3(b) holds. Then,

$$
\begin{aligned}
& \alpha l(\mathbf{y}, \theta_1) + \beta l(\mathbf{y}, \theta_2) - l(\mathbf{y}, \alpha\theta_1 + \beta\theta_2) \\
=\ & \alpha\|\mathbf{y} - \mathbf{x}(\theta_1)\|_2^2 + \beta\|\mathbf{y} - \mathbf{x}(\theta_2)\|_2^2 - \|\mathbf{y} - \mathbf{x}(\alpha\theta_1 + \beta\theta_2)\|_2^2 \\
=\ & \alpha\|\mathbf{y} - \mathbf{x}(\theta_1)\|_2^2 + \beta\|\mathbf{y} - \mathbf{x}(\theta_2)\|_2^2 - \|\mathbf{y} - \alpha\mathbf{x}(\theta_1) - \beta\mathbf{x}(\theta_2)\|_2^2 \\
& + \|\mathbf{y} - \alpha\mathbf{x}(\theta_1) - \beta\mathbf{x}(\theta_2)\|_2^2 - \|\mathbf{y} - \mathbf{x}(\alpha\theta_1 + \beta\theta_2)\|_2^2 \\
=\ & \alpha\beta\|\mathbf{x}(\theta_1) - \mathbf{x}(\theta_2)\|_2^2 + \|\mathbf{y} - \alpha\mathbf{x}(\theta_1) - \beta\mathbf{x}(\theta_2)\|_2^2 - \|\mathbf{y} - \mathbf{x}(\alpha\theta_1 + \beta\theta_2)\|_2^2 \\
=\ & \alpha\beta\|\mathbf{x}(\theta_1) - \mathbf{x}(\theta_2)\|_2^2 - \langle \alpha\mathbf{x}(\theta_1) + \beta\mathbf{x}(\theta_2) - \mathbf{x}(\alpha\theta_1 + \beta\theta_2), 2\mathbf{y} - \mathbf{x}(\alpha\theta_1 + \beta\theta_2) - \alpha\mathbf{x}(\theta_1) - \beta\mathbf{x}(\theta_2)\rangle \\
\geq\ & \alpha\beta\|\mathbf{x}(\theta_1) - \mathbf{x}(\theta_2)\|_2^2 - \|\alpha\mathbf{x}(\theta_1) + \beta\mathbf{x}(\theta_2) - \mathbf{x}(\alpha\theta_1 + \beta\theta_2)\|_2 \|2\mathbf{y} - \mathbf{x}(\alpha\theta_1 + \beta\theta_2) - \alpha\mathbf{x}(\theta_1) - \beta\mathbf{x}(\theta_2)\|_2
\end{aligned}
\tag{10}
$$

The last inequality is by Cauchy-Schwartz inequality. Note that

$$
\begin{aligned}
& \|\alpha\mathbf{x}(\theta_1) + \beta\mathbf{x}(\theta_2) - \mathbf{x}(\alpha\theta_1 + \beta\theta_2)\|_2 \|2\mathbf{y} - \mathbf{x}(\alpha\theta_1 + \beta\theta_2) - \alpha\mathbf{x}(\theta_1) - \beta\mathbf{x}(\theta_2)\|_2 \\
\leq\ & 2(B + R)\|\alpha\mathbf{x}(\theta_1) + \beta\mathbf{x}(\theta_2) - \mathbf{x}(\alpha\theta_1 + \beta\theta_2)\|_2 \\
\leq\ & \alpha\beta\|\mathbf{x}(\theta_1) - \mathbf{x}(\theta_2)\|_2 \quad \text{(By Assumption 3.3(b))}
\end{aligned}
\tag{11}
$$

Plugging equation 11 in equation 10 yields the result.

Using Theorem 3.2. in Kulis & Bartlett (2010), for $\alpha_t \leq \frac{G_t(\theta_{t+1})}{G_t(\theta_t)}$, we have

$$
\begin{aligned}
R_T \leq\ & \sum_{t=1}^{T} \frac{1}{\eta_t}(1 - \alpha_t)\eta_t l(\mathbf{y}_t, \theta_t) \\
& + \frac{1}{2\eta_t}(\|\theta_t - \theta^*\|_2^2 - \|\theta_{t+1} - \theta^*\|_2^2)
\end{aligned}
\tag{12}
$$

Notice that

$$
\begin{aligned}
& G_t(\theta_t) - G_t(\theta_{t+1}) \\
=\ & \eta_t(l(\mathbf{y}_t, \theta_t) - l(\mathbf{y}_t, \theta_{t+1})) - \frac{1}{2}\|\theta_t - \theta_{t+1}\|_2^2 \\
\leq\ & \frac{4(B+R)\kappa\eta_t}{\lambda}\|\theta_t - \theta_{t+1}\|_2 - \frac{1}{2}\|\theta_t - \theta_{t+1}\|_2^2 \\
\leq\ & \frac{8(B+R)^2\kappa^2\eta_t^2}{\lambda^2}
\end{aligned}
\tag{13}
$$

The first inequality follows by applying Lemma 3.1.

Let $\alpha_t = \frac{R_t(\theta_{t+1})}{R_t(\theta_t)}$. Using equation 13, we have

$$
\begin{aligned}
(1 - \alpha_t)\eta_t l(\mathbf{y}_t, \theta_t) & = (1 - \alpha_t)G_t(\theta_t) \\
& = G_t(\theta_t) - G_t(\theta_{t+1}) \\
& \leq \frac{8(B+R)^2\kappa^2\eta_t^2}{\lambda^2}
\end{aligned}
\tag{14}
$$

Plug equation 14 in equation 12, and note the telescoping sum,

$$
R_T \leq \sum_{t=1}^{T} \frac{8(B+R)^2\kappa^2\eta_t}{\lambda^2}
$$

$$
+ \sum_{t=1}^{T} \frac{1}{2\eta_t}(\|\theta_t - \theta^*\|_2^2 - \|\theta_{t+1} - \theta^*\|_2^2)
$$

Setting $\eta_t = \frac{D\lambda}{2(B+R)\kappa\sqrt{2t}}$, we can simplify the second summation to $\frac{D(B+R)\kappa\sqrt{2}}{\lambda}$ since the sum telescopes and $\theta_1 = \mathbf{0}, \|\theta^*\|_2 \leq D$. The first sum simplifies using $\sum_{t=1}^T \frac{1}{\sqrt{t}} \leq 2\sqrt{T} - 1$ to obtain the result

$$R_T \leq \frac{4\sqrt{2}(B+R)D\kappa}{\lambda}\sqrt{T}.$$

$\square$

### A.3 OMITTED EXAMPLES

#### A.3.1 EXAMPLES FOR WHICH ASSUMPTION 3.3 HOLDS

Consider for example the following quadratic program

$$\min_{\mathbf{x}\in\mathbb{R}^n} \quad \begin{pmatrix} \mathbf{x}^T\mathbf{x} - 2\theta_1^T\mathbf{x} \\ \mathbf{x}^T\mathbf{x} - 2\theta_2^T\mathbf{x} \end{pmatrix}$$

$$s.t. \quad 0 \leq \mathbf{x} \leq 10$$

One can check that Assumption 3.3 (a) is indeed satisfied. For example, let $n = 1$. Then, W.L.O.G, let $\theta_1 \leq \theta_2$. Then, $X_E(\theta) = [\theta_1, \theta_2]$. Consider two parameters that $\theta^1 = (\theta_1^1, \theta_2^1), \theta^2 = (\theta_1^2, \theta_2^2) \in [0, 10]^2$. For all $\alpha \in [0, 1]$,

$$X_E(\alpha\theta^1 + (1-\alpha)\theta^2) = [\alpha\theta_1^1 + (1-\alpha)\theta_1^2, \alpha\theta_2^1 + (1-\alpha)\theta_2^2]$$

Although tedious, one can check that one can check that Assumption 3.3 (a) is indeed satisfied.

### A.4 DATA FOR THE PORTFOLIO OPTIMIZATION PROBLEM

Table 2: True Expected Return

| Security | 1 | 2 | 3 | 4 | 5 | 6 | 7 | 8 |
|---|---|---|---|---|---|---|---|---|
| Expected Return | 0.1791 | 0.1143 | 0.1357 | 0.0837 | 0.1653 | 0.1808 | 0.0352 | 0.0368 |

Table 3: True Return Covariances Matrix

| Security | 1 | 2 | 3 | 4 | 5 | 6 | 7 | 8 |
|---|---|---|---|---|---|---|---|---|
| 1 | 0.1641 | 0.0299 | 0.0478 | 0.0491 | 0.058 | 0.0871 | 0.0603 | 0.0492 |
| 2 | 0.0299 | 0.0720 | 0.0511 | 0.0287 | 0.0527 | 0.0297 | 0.0291 | 0.0326 |
| 3 | 0.0478 | 0.0511 | 0.0794 | 0.0498 | 0.0664 | 0.0479 | 0.0395 | 0.0523 |
| 4 | 0.0491 | 0.0287 | 0.0498 | 0.1148 | 0.0336 | 0.0503 | 0.0326 | 0.0447 |
| 5 | 0.0580 | 0.0527 | 0.0664 | 0.0336 | 0.1073 | 0.0483 | 0.0402 | 0.0533 |
| 6 | 0.0871 | 0.0297 | 0.0479 | 0.0503 | 0.0483 | 0.1134 | 0.0591 | 0.0387 |
| 7 | 0.0603 | 0.0291 | 0.0395 | 0.0326 | 0.0402 | 0.0591 | 0.0704 | 0.0244 |
| 8 | 0.0492 | 0.0326 | 0.0523 | 0.0447 | 0.0533 | 0.0387 | 0.0244 | 0.1028 |

### A.5 APPROXIMATION ERROR

**Theorem A.1.** Under Assumption 3.1, we have that $\forall \mathbf{y} \in \mathcal{Y}, \forall \theta \in \Theta$,

$$0 \leq l_K(\mathbf{y}, \theta) - l(\mathbf{y}, \theta) \leq \frac{4(B+R)\zeta}{\lambda} \cdot \frac{\sqrt{2p}}{\Lambda - 1},$$

where

$$K = \frac{(\Lambda + p - 2)!}{(\Lambda - 1)!(p - 1)!}, \zeta = \max_{l \in [p], \mathbf{x} \in X(\theta), \theta \in \Theta} |f_l(\mathbf{x}, \theta)|.$$

Furthermore,

$$0 \leq l_K(\mathbf{y}, \theta) - l(\mathbf{y}, \theta) \leq \frac{16e(B+R)\zeta}{\lambda} \cdot \frac{1}{K^{\frac{1}{p-1}}}.$$

Thus, the surrogate loss function uniformly converges to the loss function at the rate of $\mathcal{O}(1/K^{\frac{1}{p-1}})$. Note that this rate exhibits a dependence on the number of objective functions $p$. As $p$ increases, we might require (approximately) exponentially more weight samples $\{w_K\}_{k\in[K]}$ to achieve an approximation accuracy. In fact, this phenomenon is a reflection of *curse of dimensionality* Hastie et al. (2001), a principle that estimation becomes exponentially harder as the number of dimension increases. In particular, the dimension here is the number of objective functions $p$. Naturally, one way to deal with the curse of dimensionality is to employ dimension reduction techniques in statistics to find a low-dimensional representation of the objective functions.

**Example A.1.** When $p = 2$, MOP is a bi-objective decision making problem. Then, Theorem A.1 shows that $l_K(\mathbf{y}, \theta) - l(\mathbf{y}, \theta)$ is of $\mathcal{O}(1/K)$. That is, $l_K(\mathbf{y}, \theta)$ asymptotically converges to $l(\mathbf{y}, \theta)$ sublinearly.

*Proof.* By definition,

$$l_K(\mathbf{y}, \theta) - l(\mathbf{y}, \theta) = \min_{\mathbf{x} \in \bigcup_{k \in [K]} S(w_k, \theta)} \|\mathbf{y} - \mathbf{x}\|_2^2 - \min_{\mathbf{x} \in X_E(\theta)} \|\mathbf{y} - \mathbf{x}\|_2^2 \geq 0.$$

Let $\|\mathbf{y} - S(w_k^{\mathbf{y}}, \theta)\|_2^2 = \min_{\mathbf{x} \in \bigcup_{k \in [K]} S(w_k, \theta)} \|\mathbf{y} - \mathbf{x}\|_2^2$, and $\|\mathbf{y} - S(w^{\mathbf{y}}, \theta)\|_2^2 = \min_{\mathbf{x} \in X_E(\theta)} \|\mathbf{y} - \mathbf{x}\|_2^2$. Let $w_{k'}^{\mathbf{y}}$ be the closest weight sample among $\{w_k\}_{k \in [K]}$ to $w^{\mathbf{y}}$. Then,

$$
\begin{aligned}
l_K(\mathbf{y}, \theta) - l(\mathbf{y}, \theta) &= \|\mathbf{y} - S(w_k^{\mathbf{y}}, \theta)\|_2^2 - \|\mathbf{y} - S(w^{\mathbf{y}}, \theta)\|_2^2 \\
&\leq \|\mathbf{y} - S(w_{k'}^{\mathbf{y}}, \theta)\|_2^2 - \|\mathbf{y} - S(w^{\mathbf{y}}, \theta)\|_2^2 \\
&= (2\mathbf{y} - S(w_{k'}^{\mathbf{y}}, \theta) - S(w^{\mathbf{y}}, \theta))^T (S(w^{\mathbf{y}}, \theta) - S(w_{k'}^{\mathbf{y}}, \theta)) \\
&\leq \|2\mathbf{y} - S(w_{k'}^{\mathbf{y}}, \theta) - S(w^{\mathbf{y}}, \theta)\|_2 \|S(w^{\mathbf{y}}, \theta) - S(w_{k'}^{\mathbf{y}}, \theta)\|_2 \\
&\leq 2(B + R)\|S(w^{\mathbf{y}}, \theta) - S(w_{k'}^{\mathbf{y}}, \theta)\|_2 \\
&\leq \frac{4(B+R)\zeta\sqrt{p}}{\lambda} \cdot \|w^{\mathbf{y}} - w_{k'}^{\mathbf{y}}\|_2,
\end{aligned}
\tag{15}
$$

where $\zeta = \max_{l \in [p], \mathbf{x} \in X(\theta), \theta \in \Theta} |f_l(\mathbf{x}, \theta)|$. The third inequality is due to Cauchy Schwarz inequality. Under Assumption 3.1, we can apply Lemma 4 in Dong & Zeng (2018) to yield the last inequality.

Next, we will show that $\forall w \in \mathscr{W}_p$, the distance between $w$ and its closest weight sample among $\{w_k\}_{k \in [K]}$ is upper bounded by the function of $K$ and $p$ and nothing else. More precisely, we will show that

$$\sup_{w \in \mathscr{W}_p} \min_{k \in [K]} \|w - w_k\|_2 \leq \frac{\sqrt{2}}{\Lambda - 1}. \tag{16}$$

Here, $\Lambda$ is the number of evenly spaced weight samples between any two extreme points of $\mathscr{W}_p$.

Note that $\{w_k\}_{k \in [K]}$ are evenly sampled from $\mathscr{W}_p$, and that the distance between any two extreme points of $\mathscr{W}_p$ equals to $\sqrt{2}$. Hence, the distances between any two neighboring weight samples are equal and can be calculated as the distance between any two extreme points of $\mathscr{W}_p$ divided by $\Lambda - 1$. Proof of equation 16 can be done by further noticing that the distance between any $w$ and $\{w_k\}_{k \in [K]}$ is upper bounded by the distances between any two neighboring weight samples.

Combining equation 15 and equation 16 yields that

$$0 \leq l_K(\mathbf{y}, \theta) - l(\mathbf{y}, \theta) \leq \frac{4(B+R)\zeta}{\lambda} \cdot \frac{\sqrt{2p}}{\Lambda - 1}, \tag{17}$$

Then, we can prove that the total number of weight samples $K$ and $\Lambda$ has the following relationship:

$$K = \binom{\Lambda + p - 2}{p - 1} \tag{18}$$

Proof of equation 18 can be done by induction with respect to $p$. Obviously, equation 18 holds when $p = 2$ as $K = \Lambda$. Assume equation 18 holds for the $\leq p - 1$ cases. For ease of notation, denote

$$K_p^\Lambda = \binom{\Lambda + p - 2}{p - 1}.$$

Then, for the $p$ case, we note that the weight samples can be classified into two categories: $w_p = 0; w_p > 0$. For $w_p = 0$, the number of weight samples is simply $K_{p-1}^\Lambda$. For $w_p > 0$, the number of weight samples is $K_p^{\Lambda-1}$. Thus,

$$K = K_{p-1}^\Lambda + K_p^{\Lambda-1}. \tag{19}$$

Iteratively expanding $K_p^{\Lambda-1}$ through the same argument as equation 18 and using the fact that

$$\binom{n}{k} = \binom{n-1}{k-1} + \binom{n-1}{k},$$

we have

$$
\begin{aligned}
K \quad &= K_{p-1}^\Lambda + K_p^{\Lambda-1} = K_{p-1}^\Lambda + K_{p-1}^{\Lambda-1} + K_p^{\Lambda-2} \\
&\vdots \\
&= K_{p-1}^\Lambda + K_{p-1}^{\Lambda-1} + \cdots + K_{p-1}^2 + K_p^1 \\
&= \binom{\Lambda+p-3}{p-2} + \binom{\Lambda+p-4}{p-2} + \cdots + \binom{p-1}{p-2} + \binom{p-1}{p-1} \\
&= \frac{(\Lambda+p-2)!}{(\Lambda-1)!(p-1)!}
\end{aligned}
\tag{20}
$$

To this end, we complete the proof of equation 18.

Furthermore, we notice that

$$K = \frac{(\Lambda+p-2)!}{(\Lambda-1)!(p-1)!} \leq \frac{(\Lambda+p-2)^{p-1}}{(p-1)!} < \left(\frac{\Lambda+p-2}{p-1}\right)^{p-1} \cdot e^{p-1}.$$

Then, when $\Lambda \geq p(K \geq 2^{p-1})$, through simple algebraic calculation we have

$$\frac{e}{K^{\frac{1}{p-1}}} > \frac{p-1}{\Lambda+p-2} > \frac{1}{4} \cdot \frac{p}{\Lambda-1} \tag{21}$$

We complete the proof by combining equation 17 and equation 21 and noticing that $\sqrt{2p} \leq p$. $\quad\square$

