# OpenReview forum: "Learning Multiobjective Program Through Online Learning"
_ICLR.cc/2023/Conference — Submitted to ICLR 2023_

### Official Review · Reviewer_4Sxk · 2022-10-19

**Confidence:** 3
**Correctness:** 3
**Technical Novelty And Significance:** 3
**Empirical Novelty And Significance:** 3
**Recommendation:** 5

**Clarity, Quality, Novelty And Reproducibility:**

Clarity: there are many sections where the writing could be improved. See also strengths and weaknesses. I found the math reasonably clear (but did not check all details).
- Use parenthetical citations (\citep) instead of \citet—see the instructions in the template
- The phrase “actually carries the data-driven concept” appears several times and I don’t know what it means
- Bottom of pg. 3—loss lunction
- I didn’t follow the last sentence of Remark 2.1—what is strict convexity buying us here?
- Put the \ref in parentheses when referring to equations (or write Eq.~\ref{})
- Assumption 3.1 “relatively interior”
- Sec 4 1st para “Gurobi Gurobi”

Quality: see strengths and weaknesses. The work solves a problem, but it's more of a potential building block for a larger problem.

Novelty: the work builds on past work, but from a skim of past work, appears sufficiently distinct.

Reproducibility: the algorithms are described sufficiently. I do not see details about a code release.

**Strength And Weaknesses:**

Strengths:
- The setting and algorithm appear novel.
- The experiments provide a nice evaluation of the two variants of the algorithm.

Weaknesses:
- The alignment between the work and motivation seems a bit off. The most natural motivation from my perspective is being able to predict the decision better under unseen settings where we have some salient information, such as new estimated returns and risks in the portfolio optimization setting. As it stands, nothing appears to change between observed decisions except the realized noise—so there doesn't seem to be a strong reason to infer the parameter vector $\theta$.
- The authors also gesture towards suboptimality of the observed decisions, which seems like a relevant problem, but their model doesn't really accommodate this because the noise is simply added to the decisions themselves. I was expecting to see noise added farther upstream—either to the objective functions themselves or to the parameters $\theta$.
- The sequence of intro to related work is poorly written, in my opinion, starting from about the second paragraph. The end of the intro serves as a mini related work section on its own, but it doesn't make clear what the authors' contributions are in detail or directly clarify what is different from past work.

**Summary Of The Paper:**

The authors study an inverse online multiobjective problem. In particular, the learner noisily observes a sequence of decisions made by a decision-maker. The learner's goal is to infer the parameters of the multiple objectives used by the decision-maker.

The authors give an online learning algorithm for this setting that achieves $1/\sqrt{T}$ regret under some regularity conditions—namely strong convexity of the individual objectives and Lipschitzness in how the decision changes as a function of the parameters. A key issue is efficient computation of the efficient frontier and how approximation affects algorithm performance. Motivated by the theory, the authors introduce an accelerated variant.

In the experiments, the authors illustrate the estimation error of each algorithm, its computational cost and how well its estimated efficient set matches the true one.

**Summary Of The Review:**

The work takes a new incremental step that \emph{may} be in the direction of solving an interesting problem.

Post-response:
I am somewhat convinced by the authors' response but not fully. (The updated version also fails to fix many of the small presentational issues that the paper has).

---

### Official Review · Reviewer_SLew · 2022-10-20

**Confidence:** 3
**Correctness:** 4
**Technical Novelty And Significance:** 2
**Empirical Novelty And Significance:** 2
**Recommendation:** 5

**Clarity, Quality, Novelty And Reproducibility:**

This paper is well-written and easy to follow. The theoretical analysis seems to be correct, and enough details of experiments are provided. However, given the existing work of Dong et al. (2018), the novelty of this paper is limited.

Moreover, there are some suggestions.
1) The caption of Figure 1 has some typos. For example, we cannot find Figure 1(e).
2) In Figure 1, the authors only compare the running time between the batch setting and online setting. The authors may also compare the error between the batch setting and online setting.
3) In Figure 1(b), the run time of Alg. 1 and Alg. 2 are stacked, which is not convenient for comparison.

**Strength And Weaknesses:**

#Strength
1) The problem of inverse multiobjective optimization is interesting.
2) The authors propose an ideal online implicit update method with the $O(\sqrt{T})$ regret bound, and provide two efficient variants of this ideal method.

#Weaknesses
1) Dong et al. (2018) have studied inverse optimization via online learning and proposed an online method with $O(\sqrt{T})$ regret bound. Although this paper further considers the inverse multiobjective optimization, which is more general than Dong et al. (2018), both the ideal online implicit update method and the regret bound are very similar to those in Dong et al. (2018), which limits the novelty of this paper.
2) Compared with Dong et al. (2018), the additional challenge faced by this paper is caused by the non-existence of the closed form of the efficient set. The authors address this challenge by approximating the efficient set with a sampling approach. However, the authors do not provide theoretical guarantees about the approximation error.

**Summary Of The Paper:**

This paper studies the problem of inverse multiobjective optimization via online learning. Under some assumptions, the authors prove that an ideal online implicit update method has the regret bound of $O(\sqrt{T})$. However, this ideal method cannot be implemented in practice. To address this problem, the authors proposed two approximate variants of this ideal method, which are heuristic.

**Summary Of The Review:**

From the above comments, I think this paper is marginally below the acceptance threshold.

---

### Official Review · Reviewer_QnFE · 2022-10-25

**Confidence:** 2
**Correctness:** 3
**Technical Novelty And Significance:** 4
**Empirical Novelty And Significance:** Not applicable
**Recommendation:** 6

**Clarity, Quality, Novelty And Reproducibility:**

clarity : it appears good -- I can follow everything until 2.2.1

novelty : the authors claims that nobody has worked on the problem of inferring parameters for the form min(f1(x,theta1)...) before, so this work is novel.

quality : appears to be good

**Strength And Weaknesses:**

strength : It tackles a new problem of multi-objective reward learning, of a particular form: min(f1(x,\theta) ... fn(x,\theta)). The paper was able to take this special form, along with some assumptions, to develop a set of update-rules to guess what \theta can be as online observations come in. Even if the user (like me) does not grasp the full derivation, this algorithm can easily exist in a package to be used as black box.

weakness : for me the paper is easy to follow wherever it is self-contained, but then some thm or corr. was invoked, and this "jump" makes the paper hard to follow. however this is unlikely to be an issue with people familiar with the field.

I do have some questions, which would be great to have answered:

1. when we say the agent is minimizing regret, does it mean it is sufficient agent comes up with a hypothesis that explains the data as well as the original \theta, rather than trying to infer the original \theta itself?

2. I always find the connection between bandit-like literatures and bayesian inference related, yet the two use very different language. For me, I'd think we just put some prior over the hypothesis P(\theta), and we do posterior inference of P(\theta | Data) while assuming some forward data generating process P(Data | \theta). Or more simply, we can find the maximum-likelihood estimator without the full posterior. There must be a good reason why this kind of language isn't the way to explain your work, but what is it?

3. How would the problem change if you are allowed to take active samples? i.e. present the agent with a scenario as a query, and see how it respond to it in the style of active learning. Can your approach be made to work in this setting? How would you select a query? How would you select a query without making some distributional assumptions of the objective function, other than it is a min of a bunch of functions ?

**Summary Of The Paper:**

This paper is very technical and I'll first say I am not familiar with the literature. However, as somewhat an "outsider" to this problem, hopefully I can contribute by asking the right questions, which can help the author make this work more accessible.

My naive understanding of this paper is that it's basically doing reward learning (or inverse reinforcement learning). Given a black box agent with some latent reward function (of a particular form, that of min of bunch of other functions), we can observe the agent's actions to infer what the reward function is. The observations come sequentially, but we do not get to control what observation to make.

**Summary Of The Review:**

from a non-expert point of view, this paper can result in an artifact that can exist in scipy as a black-box, which is definitely a contribution. I would defer technical correctness to an expert.

after discussion with other reviewers, who knows the work better, I'm lowering my score to a 6.
I still think this work is interesting though.

---

### Decision · Program_Chairs · 2023-01-20

**Decision:**

Reject

**Justification For Why Not Higher Score:**

Results are too incremental.

**Justification For Why Not Lower Score:**

N/A

**Metareview: Summary, Strengths And Weaknesses:**

The paper presents an online learning approach to learn the parameter $\theta$ of a multiobjective minimization problem $\min_x [f_1(x,\theta),\ldots,f_p(x,\theta)]$ for strongly convex functions $f_i(\cdot,\theta)$, given a sequence of noisy observations $(y_t)$ for the solution $x$ from an expert. A regret bound for the loss measuring the distance of $y_t$ and the optimal solution $x$ for the estimated parameter value $\theta_t$ is proved for an idealized algorithm under some assumptions on the problem structure, and heuristic approximation algorithms (analyzed half way) are also suggested.

The paper heavily builds on the work of Dong et al. (2018), who considered the case of a single objective (i.e., $p=1$). In fact, the difference is that while the loss function mentioned above is smooth and convex in Dong et al.'s work, the current paper makes some assumptions (most importantly Assumption 3.3) on the problem structure to be able to prove this in the multi-objective case ($p>1$). However, these assumptions are unjustified (only a toy example is provided, and even there it is only argued that the first part of Assumption 3.3 holds), hence the contribution of the paper is too incremental for a publication at ICLR.


**Summary Of Ac-Reviewer Meeting:**

Although the scores may indicate that this was a borderline paper, the only (slightly) positive reviewer has low confidence and admittedly did not look into the technical contributions in depth, which is the basis of the rejection decision. The other two reviewers (and also myself) agreed that the contributions were too incremental, and the positive reviewer was also ok with this decision.